# Investigations of the Potential Application of *k*-out-of-*n* Systems in Oil and Gas Industry Objects

**Vladimir V. Rykov** [1,2,*] **, Mikhail G. Sukharev** [1,*] **and Victor Yu. Itkin** [1,*]

[1]  Dep. Applied Mathematics and Computer Modelling, National University of Oil and Gas "Gubkin University", 65 Leninsky Prospekt, 119991 Moscow, Russia

[2]  Dep. of Applied Informatics and Probability Theory, Peoples' Friendship University of Russia (RUDN University), 6 Miklukho-Maklaya str., 117198 Moscow, Russia

\*  Correspondence: vladimir_rykov@mail.ru (V.V.R.); mgsukharev@mail.ru (M.G.S.); itkin.v@gubkin.ru (V.Y.I.)

**Abstract:** The purpose of this paper was to demonstrate the possibilities of assessing the reliability of oil and gas industry structures with the help of mathematical models of *k*-out-of-*n* systems. We show how the reliability of various structures in the oil and gas complex can be described and investigated using *k*-out-of-*n* models. Because the initial information about the life and repair time of components of systems is only usually known on the scale of one and/or two moments, we focus on the problem of the sensitivity analysis of the system reliability indices to the shape of its components repair time distributions. To address this problem, we used the so-called markovization method, based on the introduction of supplementary variables, to model the system behavior with the help of the two-dimensional Markov process with discrete-continuous states. On the basis of the forward Kolmogorov equations for the time-dependent process' state probabilities, relevant balance equations for the process' stationary probabilities are presented. Using these equations, stationary probabilities and some reliability indices for two examples from the oil and gas industry were calculated and their sensitivity to the system component's repair time distributions was analyzed. Calculations show that under "rare" component failures, most system reliability indices become practically insensitive to the shape of the components repair time distributions.

**Keywords:** *k*-out-of-*n*-type systems; reliability; probability of system failure; failure-free time; repair time; stationary mode

## 1. Introduction and Motivation

The purpose of this work was to highlight the *k*-out-of-*n* model for analyzing the reliability of oil and gas facilities. We demonstrate the capabilities of this model using two examples of offshore and onshore structures of the oil and gas complex. Hydrocarbons are successfully produced in the seas and oceans both on the shelf and in deep water. The scale of production is steadily expanding. The extracted raw materials are transported to places of consumption on oil tankers and methane carriers, but the main mode of transport is pipelines. When designing large-scale facilities for oil and gas pipeline transportation systems, and developing plans for their operation and development in the medium and long term, it is always envisaged to reserve production capacities. This is due to the importance of hydrocarbons in modern energetics and economics. According to Russian [1] and international standards, there are various types of redundancy, among which one of the most common is structural redundancy. Structural reservation is the use of redundant structural elements. While structural reservation at pumping and compressor stations is standard practice, structural redundancy at the line section is relatively rare. Reservation of pipelines is provided when the route crosses high-risk water barriers, mainly rivers. Some rivers are characterized by natural phenomena accompanied by

increases in water discharge that are difficult to predict. Therefore, crossings over problem rivers should be reserved. To increase the reliability of the pipeline system, an additional, parallel pipeline (it is called siphon barrels) is laid across the river. The diameter of the siphon barrels is less than the diameter of the main pipeline, but the total throughput of the crossing exceeds the throughput of the main pipeline. In addition, a section of the route is often reserved in dangerous mountainous areas prone to landslides, avalanches, or other natural phenomena that lead to ground movement.

Special research is usually carried out to justify reservation decisions in each specific situation. A linear, unique example of redundancy is the multi-line crossing through the Baydaratskaya Bay (Figure 1).

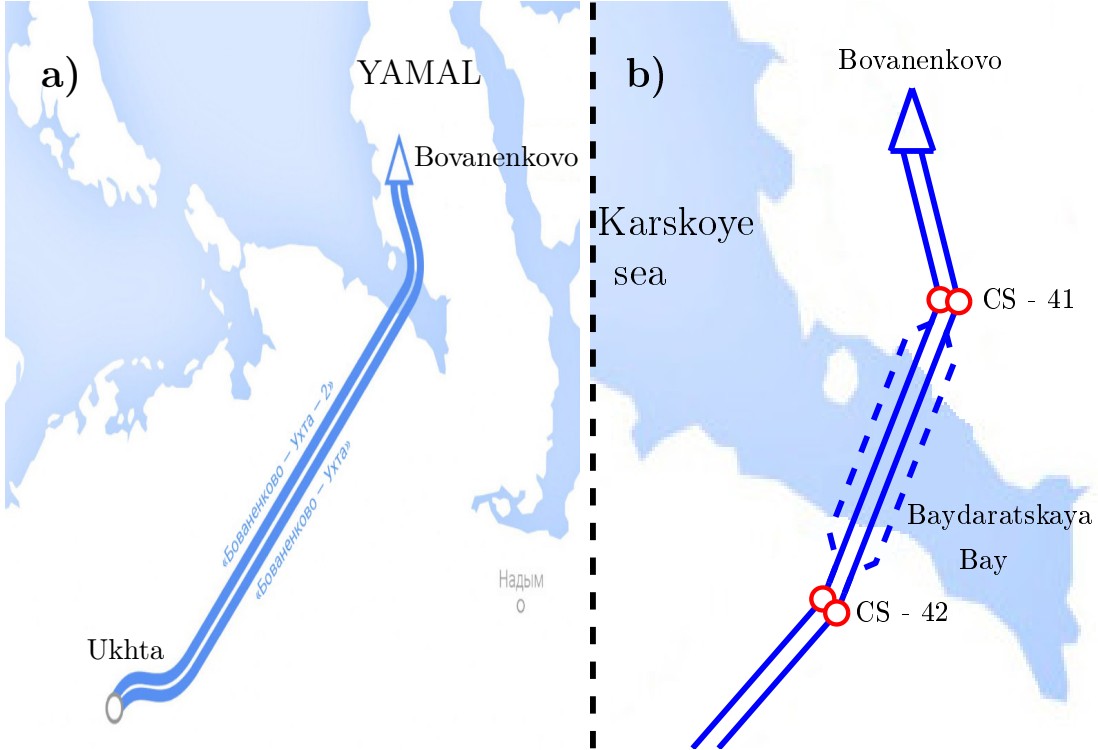

**Figure 1.** (**a**) The Bovanenkovo–Ukhta gas pipeline. (**b**) The crossing through the Baydaratskaya Bay.

The crossing is part of the technical corridor of the Bovanenkovo–Ukhta trunk gas pipelines, designed to transport gas from the Yamal Peninsula to the Unified Gas Supply System of Russia. Currently, Yamal gas production is carried out at the largest Yamal field—Bovanenkovskoye. The first stage of the corridor was commissioned in 2012, the second, in 2017.

The diameter of the main pipe is 1400 mm, the working pressure is 11.8 MPa (120 at), and the length of the underwater section is 70 km. The crossing is the most difficult section of the corridor and is made of concrete steel pipes with a diameter of 1200 mm. The development of the corridor has been outlined, including the crossing through the Baydaratskaya Bay. The transition lines included in the project, but not yet built, are shown in Figure 1b dotted line.

The creation of the Bovanenkovo–Ukhta gas transmission corridor is one of the largest and most complex projects in the history of pipeline construction. During its implementation, innovative technologies and highly reliable equipment were used. The Baydaratskaya Bay, along the bottom of which the track is laid, is covered with ice for most of the year. The bay is distinguished by special natural and climatic conditions [2]: at a shallow depth, it is characterized by frequent stormy weather, complex bottom sediments, and freezing to the bottom in winter.

When designing the corridor, it was necessary to make a decision on the structural reserve of the crossing, that is, to choose the number of lines $n$ and consider various options for emergency situations associated with the failure of $k$ lines. To justify the decision, it was necessary to consider various

situations that could potentially cause a failure and to evaluate the probabilities of such situations. Failure (here, a *failure* should be understood as a rupture to a line or damage forcing operations to halt) can occur, for example, as a result of damage caused by icebergs, which are carried into Baydaratskaya Bay by one of the Gulf Stream jets, which weaken at these latitudes.

Note that the Boolean reliability model for a bay crossing system, when the system and each of its elements are only in states 0 and 1, is not directly applicable. The gas pipeline corridor will partially fulfil its functions until all the lines of the crossing are out of order. The degree of its function performance is determined by the throughput of the entire Bovanenkovo—Ukhta gas pipeline (Figure 1a). However, it is possible to calculate this indicator by considering all possible *k*-out-of-*n* situations.

According to the authors, the possibility of using *k*-out-of-*n* models is determined by the technological specifics of the object. Therefore, the permissible number and connection schemes of the compressor shop units are often uniquely determined by the load on the pipeline *(compressor shop* is a group of compressors working in common mode; on a two-line gas pipeline, the compressor station includes two compressor shops). If the number of gas compressors is insufficient, the shop (all units of the shop) is turned off (the so-called "on-pass" mode).

In other cases, for example, when investigating the reliability of crossing a river-line obstacle, it is necessary to calculate the possibility of several *k*-out-of-*n* situations in order to find the distribution of the corridor's capacity, depending on the number of operational lines.

The problem of choosing a structural reserve was further complicated by the multi-stage construction of the corridor and crossing the bay. The life cycle of these objects is determined not only by the predicted durability of the pipelines, but also by the rational rates of gas extraction from the fields of the Yamal Peninsula in the long term. The rate of production should be tailored to the gas needs of the constituent entities of the Russian Federation, export contracts, and the schedules of possible withdrawals from the fields of the Nadym-Purtazovsky region—the main gas supplier in modern times.

The problems of the reliability of large pipeline systems for oil and gas supply was given exceptional attention during the period of their intensive development, when several thousand kilometers of pipelines were laid annually in the USSR. Various mathematical models were developed [3]. A scientific seminar "Methodological issues of researching the reliability of large energy systems" was organized that is still in effect: this year, the 92nd meeting was held. Within the framework of the seminar, a four-volume desk book on the reliability of energy systems was prepared and published. The publication reflected the state of the subject at that time in detail. A separate volume of the desk book (in two books) is devoted exclusively to the reliability of gas and oil supply systems [4,5]. Since then, special studies concerning reliability have been carried out when designing each new main pipeline, in particular for the Yamal–West corridor (1st option) [6]. Various options of the route were compared according to reliability criteria: through the Baydaratskaya Bay and bypassing the bay along the Subpolar Urals. In addition, ref. [7] is devoted to the rationale of the decisions for crossing the bay. Naturally, in applied calculations, the methodological apparatus developed by that time was used. In this paper, a model is proposed, which is particularly applicable to the study of the reliability of the crossing through the Baydaratskaya Bay. The application of this model allows one, at the design stage, to assess the longevity of an object with greater confidence.

Another example, in which it is advisable to use *k*-out-of-*n* models, is the compressor stations of the Bovanenkovo–Ukhta gas pipeline, which are located on two shores of the bay. A structural reserve of gas-compressor units is provided in each shop of the compression stations. Going through various options for *k* operable units out of *n* installed units, it is possible to fully characterize the impact of the shop on the reliability indicators of the gas pipeline corridor.

The above examples of structural redundancy in oil and gas complex facilities can be expanded by considering the production enterprises (reservation of oil wells, electric motors in the field transport system, etc.), preparation for oil and gas transportation (separators, sedimentation tanks), processing

(number of production lines), etc. The calculation of the reliability indicators of engineering systems with a structural reserve, as a rule, can be carried out using standard statements of *k*-out-of-*n* models, a brief overview of which is given below.

The redundancy technique is widely used to improve system reliability. A typical form of redundancy is a *k*-out-of-*n* configuration. A *k*-out-of-*n* ($k \leq n$) system is a repairable redundancy system that consists of *n* components in parallel, each of which can be in one of two states: operable or not operable.

The repair of the system's components is realized by a single repair unit. A *k*-out-of-*n* system may be described in two ways, depending on the definition of the parameter *k*, as follows: the parameter *k* may represent the number of components in the system that must work in order for the entire system to work, referred to as a *k*-out-of-*n*: *G* system; or *k* may represent the number of components in the system that must fail before the entire system fails, referred to as a *k*-out-of-*n*: *F* system [8].

As a result of the wide range of practical applications, a lot of papers are devoted to the study of *k*-out-of-*n* systems. The earlier investigations deal with homogeneous binary models, where each component can take only two states "UP" or "DOWN". The probabilities of the state "UP" for each component are equal. There are vast amounts of literature on such studies (see for example Trivedi [9], Chakravarthy et al. [10], and the bibliographies therein).

For heterogeneous system investigation, the method of Universal Generating Functions can be used. The basic ideas of the method were introduced by I. Ushakov in the mid 1980s [11,12]. Since then, this method has been considerably expanded and is currently very popular (see, for example, Levitin [13] and the bibliography therein).

Further investigations of such systems have been directed at the study of systems with non-exponential life and/or repair time distributions. In [14], M.S. Moustafa considered a *k*-out-of-*n* system with exponential life and an arbitrarily distributed repair time of its components with the help of the embedded Markov chains method. In addition, ref. [15] contains a detailed analysis of 2-out-of-*n* and 3-out-of-*n* systems with general repair time distributions and an evaluation of different reliability characteristics during a system's life cycle with the use of the Laplace transform to stochastic relations. In a series of papers a review to which can be found in [16], these investigations were developed with the help of the supplementary variables method firstly proposed by D. Cox [17].

These works allow to set up and develop investigations in one of the principal directions systems' reliability study—insensitivity or low sensitivity of their characteristics to life and repair times distributions of their components. These investigations are very important, because it is very difficult (or even impossible) to find sufficient reliable information about these distributions (see Section 4.1). In [18–21], using analytical and simulation methods, the sensitivity of different system reliability characteristics to the shapes of their components' life and repair time distributions was studied.

These studies show that (at least for "rare" component failures) system reliability indicators are practically insensitive to the shapes of their components life and repair time distributions. Essential dependents take place only on their two first moments. For these investigations, Markov processes with a discrete–continuous space state are used. At that, the system's time-dependent state probabilities satisfy to the Kolmogorov forward partial differential equations. In [22], this approach was used to calculate the reliability function of a *k*-out-of-*n* heterogeneous system. In [23] the general approach to solution of special kind of partial differential equations was proposed, which was used for Kolmogorov forward system of equations for time-dependent state probabilities.

The novelty of the paper and its difference from another studies on reliability of oil and gas equipments consists in:

- A study of the *k*-out-of-*n* mathematical model with arbitrary repair time distributions;
- A calculation of the steady state probabilities of this system;
- A demonstration of this model potential applications to study of reliability problems for objects in the oil and gas industry;

- A study of system reliability indicators sensitivity to the shape of system components repair time distributions.

The paper is organized as follows. In the next section, the mathematical model and some notations are presented. Section 3 is devoted to the research regarding the stationary regime of *k*-out-of-*n* systems. In Section 4, an analysis of the input information needed for the problem considered in the examples is proposed. Sections 4.2 and 4.3 consider two potential applications ("3-out-of-4" and "2-out-of-4") of these models in engineering systems of the oil and gas industry. Finally, the last section concludes the paper.

## 2. Mathematical Model

For the considering objects modelling, we use the *k*-out-of-*n*: *F* system, which consists of *n* components and fails if *k* components fail. Failed components and the whole system are repaired by a single facility. Suppose that

- The components fail according to a Poisson flow with intensity $\alpha$;
- The random repair times of components are independent identically distributed (i.i.d.) random variables (r.v.) and their common cumulative distribution function (c.d.f.) $B(t)$ is absolutely continuous with probability density function (p.d.f.) $b(t) = B'(t)$;
- After full system failure (when the system occurs in state *k*), it is repaired during a random time with absolutely continuous c.d.f. $\Gamma(t)$.

Denote by

- *j*—the number of components in the "DOWN" state;
- $E = \{j : j = \{0, 1, \ldots k\}\}$—the system's state space, where *j* denotes the number of failed components, and the state *k* is the system's "DOWN" state;
- $\lambda_j = (n - j)\alpha$—the system failure intensity in its *j*-th state;
- $\beta(x) = \frac{B'(x)}{1 - B(x)}$, $\gamma(x) = \frac{\Gamma'(x)}{1 - \Gamma(x)}$ are conditional repair densities of components and the whole system, given that elapsed repair time is *x*;
- $b = \int_0^\infty (1 - B(x)) \, dx$—mean repair time of any of the components;
- $g = \int_0^\infty (1 - \Gamma(x)) \, dx$—mean repair time of the system after its full failure.

For the study of the above system, we use the supplementary variables method as firstly proposed by D. Cox [17], which is intensively used in series of works (see, for example, [22] and the bibliography therein). To construct the appropriate Markov process, the elapsed repair time of the component under repair is introduced as a supplementary variable $X(t)$, and consider a two-dimensional random process

$$Z = \{Z(t) = (J(t), X(t)), \ t \geq 0\},$$

where for $j > 0$, value $J(t)$ represents the number of failed components at time *t*,

$$J(t) = j, \ \text{if at time } t, \text{ the system is in state } j \in E,$$

and the supplementary variable $X(t)$ denotes elapsed time, i.e., the time spent by the repair facility to repair the component. The state space of the process *Z* is

$$\mathcal{E} = \{0, (j, \mathbb{R}^+) : j = \overline{1, k}\}.$$

As a result of the supplementary variable, the process $Z$ is a Markov one. Denote its state probability (for $j = 0$) and p.d.f.'s by

$$\pi_0(t) = \mathbf{P}\{J(t) = 0\}, \quad \pi_j(t; x) = \mathbf{P}\{J(t) = j, \ X(t) = x\} \quad (j = \overline{1, k}), \tag{1}$$

and appropriate the macro-state probabilities by

$$\pi_j(t) = \mathbf{P}\{J(t) = j\} = \int_0^t \pi_j(t; x)\, dx.$$

The system's lifetime is denoted by $T$, $T = \inf\{t : \ J(t) = k\}$, and its reliability function is denoted by $R(t) = \mathbf{P}\{T > t\}$.

The time-dependent probabilities $\pi_j(t; x)$ satisfied to the Kolmogorov forward partial differential equations. This system can be obtained from appropriate system difference equations. The last one can be construct by the usual method of comparison of the system state probabilities in a small interval between two closed time points. Further, by passing to the limit when length of the interval tends to zero, the Kolmogorov system is obtained. This system was proposed in [22]. With the help of the operator

$$D = \left( \frac{\partial}{\partial t} + \frac{\partial}{\partial x} \right),$$

these equations can be written in the form

$$
\begin{aligned}
\frac{d}{dt}\pi_0(t) &= -n\alpha\pi_0(t) + \int_0^t \pi_1(t, x)\beta(x)\, dx + \int_0^t \pi_k(t, x)\gamma(x)\, dx, \\
D\pi_1(t; x) &= -((n-1)\alpha + \beta(x))\pi_1(t; x), \\
D\pi_i(t; x) &= -((n-i)\alpha + \beta(x))\pi_i(t; x) + (n-i+1)\alpha\pi_{i-1}(t; x) \\
&\quad (i = \overline{2, k-1}), \\
D\pi_k(t; x) &= -((n-k)\alpha + \gamma(x))\pi_k(t; x) + (n-k+1)\alpha\pi_{k-1}(t; x)
\end{aligned}
\tag{2}
$$

with the initial $\pi_0(0) = 1$ and boundary conditions for $k > 2$

$$
\begin{aligned}
\pi_1(t, 0) &= n\alpha_1\pi_0(t) + \int_0^t \pi_2(t; x)\beta(x)\, dx, \\
\pi_i(t, 0) &= \int_0^t \pi_{i+1}(t; x)\beta(x)\, dx \ (i = \overline{2, k-2}), \\
\pi_{k-1}(t, 0) &= \pi_k(t, 0) = 0.
\end{aligned}
\tag{3}
$$

For the solution to this kind of partial differential equations, as shown in [23], an algorithm was proposed. The algorithm allows one to find the analytic expressions for the system time dependent state probabilities in terms of their Laplace transforms. These expressions can be used for further analytic and numerical investigations of the objects considered in the paper.

For the system reliability function calculation, the same process $Z$ with absorption in the state $k$ can be used. In [22], the reliability function of the $k$-out-of-$n : F$ system in terms of its Laplace transform was found and some examples were proposed. This approach can also be used for further investigations of the appropriate objects in the oil and gas industry. Following the main idea of the paper, we focus on the calculation of the stationary system state probabilities and their applications to engineering systems in the oil and gas industry.

## 3. Stationary Regime Study

In this section, the stationary regime of the above system is considered. Note that the state 0 is a positive atom of the process $Z$ with respect to its invariant measure. Therefore, it is positively recurrent (the Harris process) and thus its limiting probabilities

$$\pi_0 = \lim_{t \to \infty} \pi_0(t), \quad \pi_j(x) = \lim_{t \to \infty} \pi_j(t, x)$$

exist and are the system steady state probabilities.

There are least two different possible ways to calculate the steady state probabilities: (a) by passing the limits in the time-dependent state probabilities from Section 2; or (b) by balancing the equations for the steady state probabilities solution. The second option is preferable because the partial differential equations solution is not a simple problem as one can see in [23].

The balanced equations also follow from equations for the time-dependent probabilities (2) by applying the derivatives with respect to the time variable to zero. As a result, they take the form

$$
\begin{aligned}
n\alpha\pi_0 &= \int_0^\infty \pi_1(x)\beta(x)\,dx + \int_0^\infty \pi_k(x)\gamma(x)\,dx, \\
\frac{d}{dx}\pi_1(x) &= -((n-1)\alpha + \beta(x))\pi_1(x), \\
\frac{d}{dx}\pi_i(x) &= -((n-i)\alpha + \beta(x))\pi_i(x) + (n-i+1)\alpha\pi_{i-1}(x) \\
&\quad (i = \overline{2, k-1}), \\
\frac{d}{dx}\pi_k(x) &= (n-k+1)\alpha\pi_{k-1}(x) - \gamma(x)\pi_k(x),
\end{aligned}
\tag{4}
$$

jointly with the boundary conditions, which, for $k = 2$, are

$$
\begin{aligned}
\pi_1(0) &= n\alpha\pi_0, \\
\pi_2(0) &= 0,
\end{aligned}
\tag{5}
$$

and, for $k > 2$, they are

$$
\begin{aligned}
\pi_1(0) &= n\alpha_1\pi_0 + \int_0^\infty \pi_2(x)\beta(x)\,dx, \\
\pi_i(0) &= \int_0^\infty \pi_{i+1}(x)\beta(x)\,dx \ (i = \overline{2, k-2}), \\
\pi_{k-1}(0) &= \pi_k(0) = 0.
\end{aligned}
\tag{6}
$$

The analytical solution of this equation, in terms of functions $\beta(x)$ and $\gamma(x)$, can be obtained with the constant variation method. The special case 3-out-of-6 was presented in [22]. However, this is too bulky and numerically demanding for real investigations. Therefore, for practical applications, a direct numerical solution is proposed.

To represent the system (4)–(6) as a linear boundary value problem, introduce the functions

$$\pi_{k+i}(x) = \int_0^x \pi_i(x)\,dx \ (i = \overline{1, k}).$$

Thus, the additional equations

$$\frac{d\pi_{k+i}(x)}{dx} = \pi_i(x) \ (i = \overline{1,k}),$$

should be added to system (4)
and the additional relations

$$\pi_{k+i}(0) = 0 \ (i = \overline{1,k}).$$

should be added to the boundary conditions (6).

In order to eliminate integrals of unknown functions from the system (4), let us integrate all equations with derivatives over the interval from 0 to $\infty$.

The left sides of the transformed equations are $\pi_i = \lim\limits_{x\to\infty} \pi_{k+i}(x)$, and the right sides contain integrals of the form $\int\limits_0^\infty \pi_i(x)\beta(x)\,dx$ and $\int\limits_0^\infty \pi_k(x)\gamma(x)\,dx$, which are present in the first equation and in the boundary conditions. Represent these expressions from the transformed system and substitute them into the boundary conditions (6). Represent the obtained system in matrix form:

$$\frac{d\vec{\pi}}{dx}(x) = U(x)\vec{\pi}(x), \tag{7}$$

$$\vec{\pi}(0) = \alpha V \vec{\pi}(\infty) + \vec{h}, \tag{8}$$

where

- $\vec{\pi}(x) = [\pi_1(x), ..., \pi_{2k}(x)]'$ is a vector of unknown functions;
- $\vec{\pi}(\infty) = \lim\limits_{x\to\infty} \vec{\pi}(x)$ is a vector of their limiting values;
- matrix $U(x) \in \mathbb{R}^{2k\times 2k}$ consists of the following non-zero components

$$
\begin{aligned}
U_{i,i}(x) &= -(n-i)\alpha - \beta(x), \\
U_{i+1,i}(x) &= (n-i)\alpha, \\
U_{k+i,i}(x) &= 1, \ i = \overline{1,k-1}, \\
U_{k,k}(x) &= -\gamma(x), \\
U_{k+k,k}(x) &= 1.
\end{aligned}
$$

The matrix $V \in \mathbb{R}^{2k\times 2k}$ consists of the following non-zero components for $k > 2$:

$$
\begin{aligned}
V_{1,k+i+1} &= -n, \\
V_{i+1,k+k-1} &= -(n-k+1), \\
V_{i+1,k+i+1} &= n-i-1, \ i = \overline{1,k-3}, \\
V_{1,k+1} &= -1, \\
V_{1,k+k-1} &= -(2n-k+1), \\
V_{1,k+k} &= -n,
\end{aligned}
$$

and for $k = 2$:

$$
\begin{aligned}
V_{1,k+k-1} &= -n, \\
V_{1,k+k} &= -n.
\end{aligned}
$$

Vector $\vec{h} \in \mathbb{R}^{2k}$ only has one non-zero component $h_1 = n\alpha$.

For example, for the 3-out-of-4 system, these matrices and vectors appear as follows:

$$U(x) \ = \ \begin{bmatrix} -3\alpha - \beta(x) & 0 & 0 & 0 & 0 & 0 \\ 3\alpha & -2\alpha - \beta(x) & 0 & 0 & 0 & 0 \\ 0 & 2\alpha & -\gamma(x) & 0 & 0 & 0 \\ 1 & 0 & 0 & 0 & 0 & 0 \\ 0 & 1 & 0 & 0 & 0 & 0 \\ 0 & 0 & 1 & 0 & 0 & 0 \end{bmatrix} ;$$

$$V \ = \ \begin{bmatrix} 0 & 0 & 0 & -1 & -6 & -4 \\ 0 & 0 & 0 & 0 & 0 & 0 \\ 0 & 0 & 0 & 0 & 0 & 0 \\ 0 & 0 & 0 & 0 & 0 & 0 \\ 0 & 0 & 0 & 0 & 0 & 0 \\ 0 & 0 & 0 & 0 & 0 & 0 \end{bmatrix} , \qquad \vec{h} = \begin{bmatrix} 4\alpha \\ 0 \\ 0 \\ 0 \\ 0 \\ 0 \end{bmatrix} .$$

For the numerical solution, it is necessary to change the infinite interval of the repair time to the finite one. Furthermore, for some distributions, the conditional probability density function is undefined if $x = 0$. Thus, it should be solved the system for $x > 0$ only. To do this, we need to choose some small parameters $p$ (for example $p = 0.001$) and solve the considered system of equations in interval $[x_p, x_{1-p}]$, where $x_p$ is the minimum of $p$-quantiles for distributions $B(x)$ and $\Gamma(x)$, and $x_{1-p}$ is the maximum of $(1 - p)$-quantiles for distributions $B(x)$ and $\Gamma(x)$.

The obtained systems (7) and (8) is a linear boundary value problem with non-separated boundary conditions. Such a problem can be solved using the standard *bvp4c* or *bvp5c* solvers in MATLAB. They implement an iterative method with three-point and four-point Lobatto formulas, respectively, which are partial cases of the implicit Runge–Kutta scheme [24]. Procedures *bvp4c* and *bvp5c* use a linear equation solver for general sparse matrices, because it is possible to solve the problem with non-separated boundary conditions (8). The solutions (7) and (8) for the special cases are given in the next section. They provide calculation any reliability indicators such as failure probability $\pi_k$ and availability $K_{av} = 1 - \pi_k$ etc.

## 4. Examples

In this section, two examples of applications of the $k$-out-of-$n$ system for study of the reliability systems in the oil and gas industry are considered.

### 4.1. Input Information for the Reliability of Engineering Systems

One of the most important and difficult problems in analyzing the reliability of complex systems in general and objects of the oil and gas industry in particular is to obtain reliable information about the reliability of their components. Complete information about the reliability of any technical object contains the distributions of two random variables: the life and repair time of all structural units of the suitable model. However, these distributions are often unknown, and one is usually limited to knowing one or two moments of these distributions: the mean and variance. In this regard, the problem of analyzing the sensitivity of the system's reliability indicators to the shape of their initial information distributions and the variability of their first moments becomes urgent. Estimates of the means and variances of life and repair time components of systems can be obtained from the statistics of the operating equipment. The main indicator of the pipeline system reliability as a whole is the rate of failures (pipe ruptures) [25]. In the 1970s, this indicator for the Unified Gas Supply System of the USSR was equal to one failure/1000 km per year. In the last decade, the failure statistics, according to the Unified Gas Supply System of Russia, indicate the stability of this parameter at the level of about 0.2 failures/1000 km per year [25]. For a 70 km section, it would be 0.014 failures/year.

Technological progress is a characteristic feature of our civilization. The technologies for the production of pipes are being improved comparatively quickly on a historical scale, and the indicators of their reliability (failure-free operation and longevity [1]) are improving. The construction and

manufacturing quality of power equipment are being improved. It is methodologically incorrect to transfer the estimators of the reliability indicators of equipment in operation to the designed facilities without making corrections for scientific and technological progress. We have to rely, to one degree or another, on expert knowledge and forecasts. For example, when assessing the reliability indicators of a unique object—the Baydaratskaya Bay crossing—it was necessary to analyze its design features and the conditions of its functioning, the natural features of the territory, and the possible causes of accidents that occur in the practice of global pipeline construction. Furthermore, it was necessary to assess the risks of accidents from non-standard situations according to the increased stability of pipes and reliability of equipment. We believe that the failure flow intensity for the lines crossing through the Baydaratskaya Bay is several times lower than the average. However, examining the sensitivity of the model, we consider options that are quite close to average intensity.

The specificity of the problems concerning the whole pipeline reliability indicator estimation consists in the need for a comparative assessment of the indicators of pipes and power equipment. There is a significant difference between failures of power equipment and pipes. The failure rate of the power equipment units is much higher than the failure rate of pipes. This can be caused by wear and tear on high-speed centrifugal blowers, the gas turbines that drive the blower, and power interruptions. According to the data from long-term operation, the operating time of gas pumping units can be considered to be obeying an exponential distribution; the average overhaul time ranges from 2000 to 5000 h, which corresponds to about 2–5 failures/year. For modern equipment, this figure should be significantly lower.

Repairing a guillotine rupture in the Baydaratskaya Bay crossing would take a very long time, which would include, in particular, moving a specialized offshore pipe layer to the polar region. The uncertainty of the repair time is aggravated by extreme climatic conditions, due to which, for example, work to restore the crossing is in principle possible only for a few months of the year. The main time spent on the repair of the line is the transportation of special equipment. Consequently, the repair would take at least a year, and in difficult cases, even more. Time expenditure depends little on the number of failed lines, i.e., it can be assumed that the distribution of the repair time of one line and the total repair time of all lines are similar.

Power equipment fails more often, but the consequences and time to eliminate the accident are usually shorter than for the linear part. In the oil and gas complex of the Russian Federation, block repairs have taken strong positions in relation to the power equipment of pipelines, i.e., the replacement of the pump, compressor, or driving engine with subsequent repair at a specialized plant. In connection with the development of new methods of repairs and repair equipment, the reliability indicators are improving. The repair of one object can take from 5 to 20 days, and the time of general repair is proportional to the number of objects.

Thus, on the basis of the analysis performed, we can conclude that:

- The failure rate of a line of Baydaratskaya Bay crossing varies from 0.004 to 0.01 failures/year;
- The failure rate of power equipment varies from 0.5 to 2 failures/year;
- The mean repair time for a line varies from 1 to 2 years;
- The mean repair time for power equipment varies from 5 to 20 days.

In conditions of a lack information, the uncertainty of estimates of reliability indicators for unique objects increases the value of robust mathematical models. In this regard, the sensitivity analysis of system reliability indicators to the distribution of its component's life and repair times is a very important. Thus, in our examples, we focus on the investigation of the sensitivity system reliability characteristics to the shapes of their components life and repair time distributions.

*4.2. Gas Pipeline through Baydaratskaya Bay*

Let us apply the mathematical apparatus described above for modeling the reliability of the pipeline system (Figure 1b) and studying the sensitivity of the model to the parameters of the distributions of time between failures and renewal time.

For the line crossing through Baydaratskaya Bay, we consider the "3-out-of-4" model, i.e., the failure of the system is considered as being the failure of three pipelines. As the main indicator of reliability, we take the stationary probability $\pi_k$ of falling into the system failure state ($\pi_k = 1 - K_{av}$).

As a result of the uncertainty of the initial information about the reliability of the elements, which was mentioned above, we consider two possible distribution laws of the repair time (the $\Gamma$-distribution and the Gnedenko–Weibull distribution) and carry out calculations for various combinations of distribution parameters.

The results are shown in Figures 2 and 3. The abscissa is the variation $V$, varying in the range from 0.1 to 2; the ordinate is the probability of system failure in the stationary mode $\pi_k$.

The graphs are plotted for different values of the failure flow intensity $\alpha$: in Figure 2, the average repair times are $b = g = 1$ year, and in Figure 3, they are $b = g = 2$ years.

For $V = 1$, both the $\Gamma$-distribution and the Gnedenko–Weibull distributions are reduced to an exponential distribution, so we can compare our calculations with the classical solution for the $k$-out-of-$n$ Markov system.

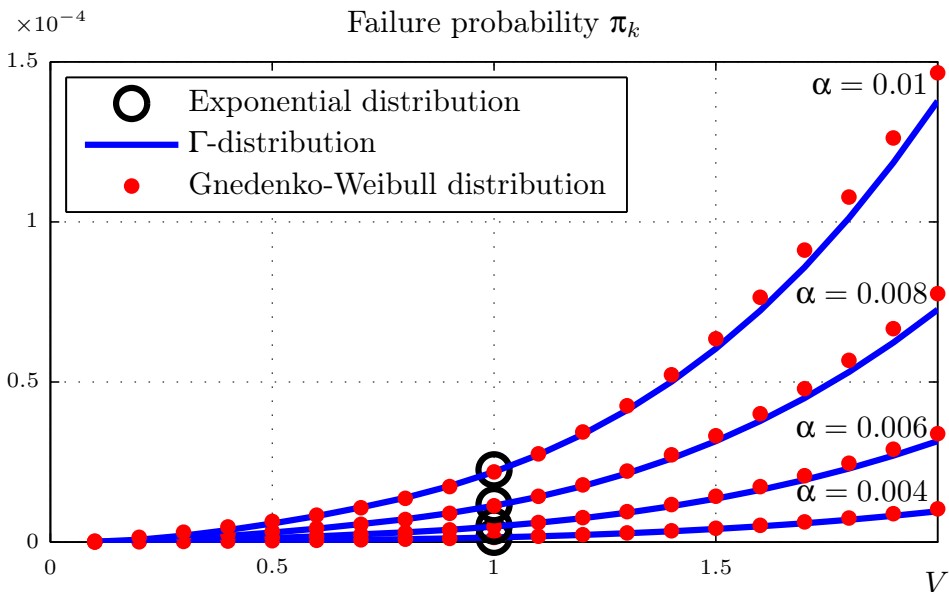

**Figure 2.** Probabilities of failure of the pipeline sections ("3-out-of-4") for $b = g = 1$ year.

Figures 2 and 3 show that there is a monotonic, but nonlinear increase in the probability of system failure with an increase in both the variation of the repair time and the failure stream intensity. The results obtained for the $\Gamma$-distribution and the Gnedenko–Weibull distribution are almost identical, which allows us to hope that the probability of failure of a line of the pipeline system is not sensitive to the type of distribution of the repair time of a line. However, the influence of the variation is quite significant. Therefore, when calculating the stationary probability of an operational state of the system (availability), the second moment of repair time of its components must be taken into account. On the other hand, the insensitivity of the system availability to the type of distribution of the repair time of its components makes it possible to use the Gnedenko–Weibull law to reduce the calculation time.

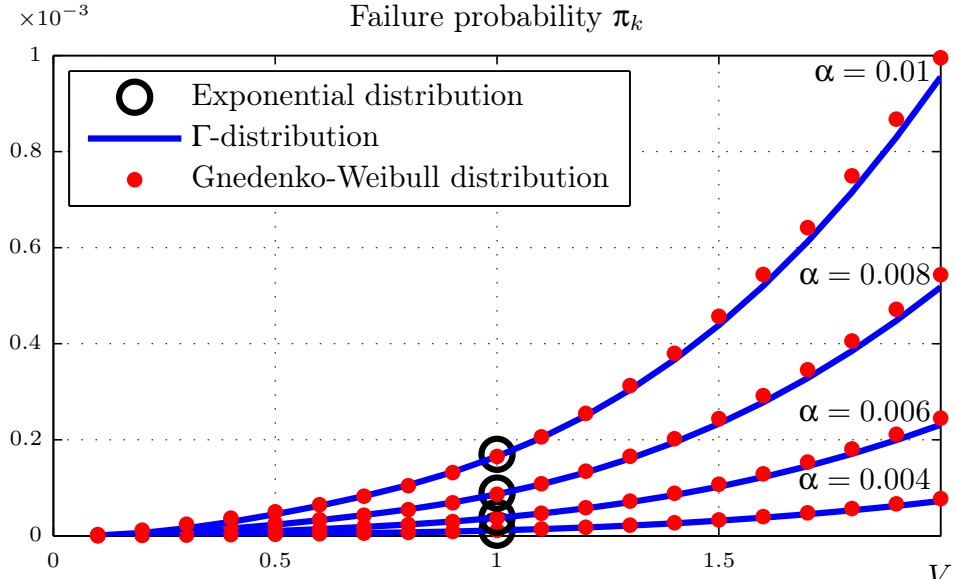

**Figure 3.** Probabilities of failure of the pipeline sections ("3-out-of-4") for $b = g = 2$ years.

### 4.3. Compressor Shop Model

Consider a model of a compressor shop equipped with $n$ gas compressors that is in operable state if fit for work are $k$ of them. We believe that the piping manifold provides moving redundancy (standby redundancy with general ratio [1]). Consider the compressor shop as a "2-out-of-4" system and carry out calculations similar to Section 4.2. The calculation results are presented in the Figures 4–6. The abscissa is the variation $V$, as before, varying in the range from 0.1 to 2; the ordinate is the probability of system failure in the stationary mode $\pi_k$. The graphs are dotted for different values of failure stream intensity $\alpha$ and mean repair times $b$.

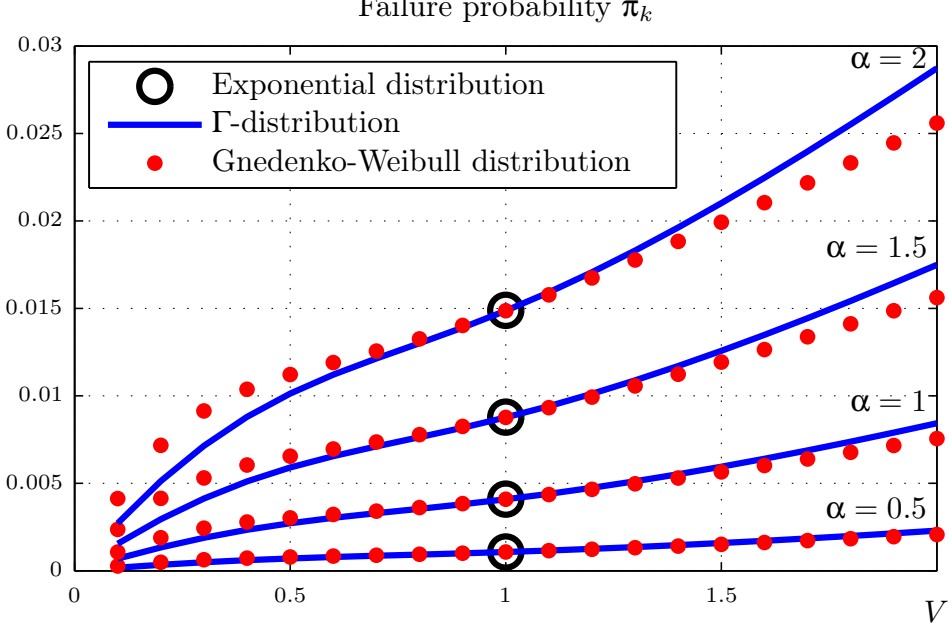

**Figure 4.** Probabilities of the compressor shop failure ("2-out-of-4") for $b = 5$ days, $g = 10$ days.

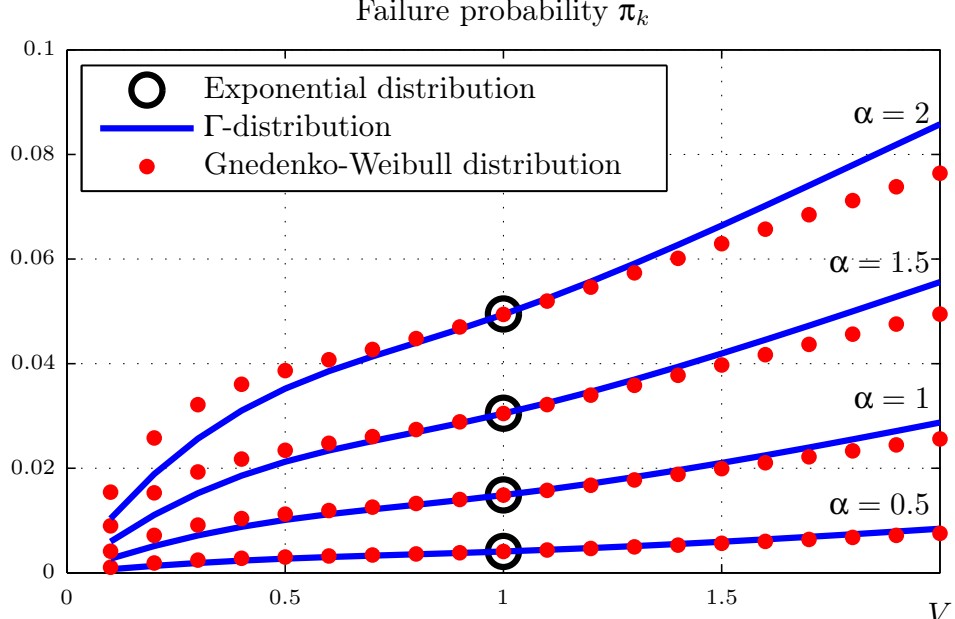

**Figure 5.** Probabilities of the compressor shop failure ("2-out-of-4") for $b = 10$ days, $g = 40$ days.

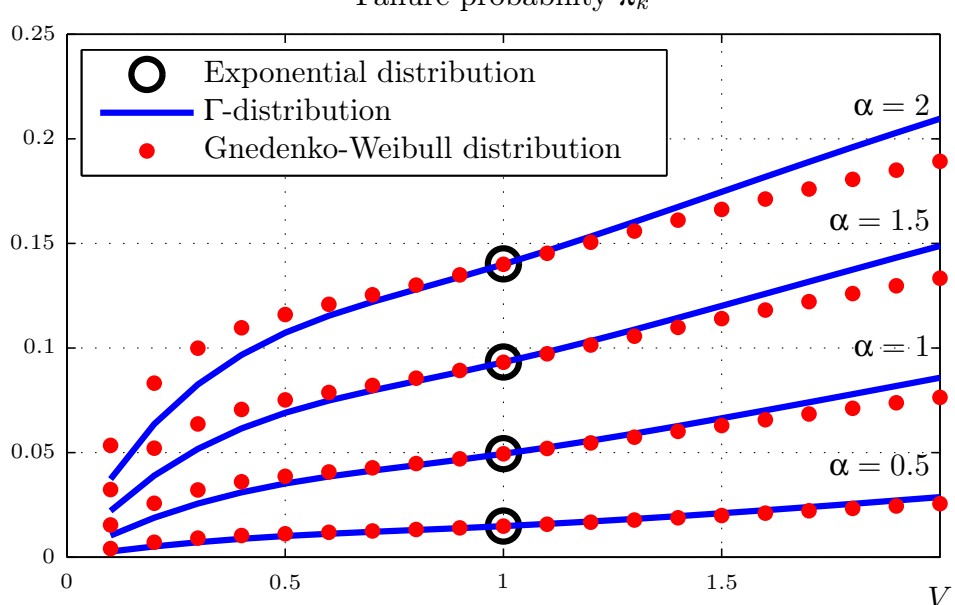

**Figure 6.** Probabilities of the compressor shop failure ("2-out-of-4") for $b = 20$ days, $g = 40$ days.

It can be seen from Figures 4–6 that in this case, the distribution law has a stronger effect on the probability of system failure, which is explained by the smaller ratio of the mean operating and repair times.

The calculations show that all the parameters of the model—mean operation time to failure and mean repair time: the repair time variation coefficient—significantly affect the reliability of the system. The shape of the repair time distribution is less important, but one should not draw far-reaching conclusions based on this study, in which only two closely related distribution laws were used.

## 5. Conclusions

The paper shows the possibility of applying the *k*-out-of-*n* model to calculating the reliability characteristics of oil and gas equipment. The practical significance of the results obtained in this work lies, first of all, in the fact that the proposed model makes it possible to reflect the conditions of the

functioning of the designed or operated object more accurately. The engineer must know how the resulting reliability indicators change depending on the initial information. This can be found out by performing a series of calculations. However, it is impossible to get an idea of the effect on the result of the assumption about the distribution law, for example, the distribution of repair time, if there are not theoretical studies. Decision makers will consider certain measures to improve the reliability justified only if there is confidence in the adequacy of the models taken into account.

Thus, the main results of the paper are as follows:

- On the basis of two examples of engineering systems in the oil and gas industry, we firstly show the possibility to calculate the reliability characteristics of corresponding systems with the help of *k*-out-of-*n* models;
- We demonstrated how the non-stationary reliability indicators of the *k*-out-of-*n* system with a Poisson flow of failures and an arbitrary distribution of repair time can be calculated by solving a system of partial differential equations for a non-stationary mode;
- An appropriate system of ordinary differential equations for a stationary regime was obtained;
- The system for stationary regime was reduced to a two-point boundary value problem, and its numerical solution was obtained with the help of suitable procedures on the MATLAB platform;
- Taking into account that complete information about a system's component's life and their repair time distributions are usually unknown, we focus on the problems of the systems reliability characteristic's sensitivity to the shape of distributions of its component's life and repair times.
- The calculations show negligible sensitivity of the system availability to their components life and repair times distributions when the ratio mean repair time to mean lifetime of each component is small. However, a system's characteristics essentially depend on the component's repair time variation.

Further investigations will focus in the proposed direction.

**Author Contributions:** Conceptualization, methodology and formal analysis, V.V.R.; setting tasks for engineering applications, compliance of terms on reliability with technical standards, M.G.S.; computer calculations, visualization and writing—original draft preparation, V.Y.I. All authors have read and agreed to the published version of the manuscript.

**Funding:** This research received no external funding.

**Acknowledgments:** The authors thanks the Publisher MDPI and the Guest Editor Pavel Praks for the permission to publish the paper for free price.

**Conflicts of Interest:** The authors declare no conflict of interest.

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
