# Peer review of "Investigations of the Potential Application of k-out-of-n Systems in Oil and Gas Industry Objects"

_jmse, doi:10.3390/jmse8110928_

Round 1

Reviewer 1 Report

the current proposal proposes to use the k-out-of-n systems properties to study Oil and Gas objects reliability. 

1- Authors need to justify the main hypothesis of the paper considering objects as a k-out-of-n systems.

2- The numerical applications for the mathematical model are clear. However, it is highly recommended to make a comparaison study with other proposals to highlight the proposed model efficiency.

3- Please give more explanation for the equations 2 and 4. 

4- Please give more details about the computation tools used for numerical results of Section 4. 

5-  The authors can make a separate section where they summarize all variables used in the paper for a clearer readability. 

Reviewer 2 Report

Recommendation: Can be accepted for review after Major revision 

The paper title suggests the relevant work in the area of oil and gas pipeline reliability modeling using the redundancy model. The authors need to focus on clarity in the work presented. Paper requires major revision and structuring. The editorial mistakes are not mentioned in the review as it's everywhere in the manuscript. The contribution of work is not clearly designed. The results need to be discussed more effectively. The literature on existing reliability models for the oil and gas pipeline should be mentioned and how the proposed approach fills the research gap should be discussed. The modeling approach is effective and the results are useful. 

Round 2

Reviewer 1 Report

The revised paper is good. The authors have considered the most of proposed corrections.

Author Response

Thanks a lot for your review.

Reviewer 2 Report

13 self-citations are observed, which is quite inappropriate. Remove unnecessary citations.

Author Response

6 self-citations are deleted. 

Thanks a lot for your review.